# Targeting the PD-1 Axis with Pembrolizumab for Recurrent or Metastatic Cancer of the Uterine Cervix: A Brief Update

**DOI:** 10.3390/ijms22041807

**Published:** 2021-02-11

**Authors:** Yannick Verhoeven, Delphine Quatannens, Xuan Bich Trinh, An Wouters, Evelien L.J. Smits, Filip Lardon, Jorrit De Waele, Peter A. van Dam

**Affiliations:** 1Center for Oncological Research (CORE), Integrated Personalized & Precision Oncology Network (IPPON), University of Antwerp, Universiteitsplein 1, B-2610 Wilrijk, Belgium; Yannick.Verhoeven@uantwerpen.be (Y.V.); Delphine.Quatannens@uantwerpen.be (D.Q.); XuanBich.Trinh@uza.be (X.B.T.); An.Wouters@uantwerpen.be (A.W.); Evelien.Smits@uza.be (E.L.J.S.); Filip.Lardon@uantwerpen.be (F.L.); Jorrit.DeWaele@uantwerpen.be (J.D.W.); 2Multidisciplinary Oncologic Centre Antwerp (MOCA), Antwerp University Hospital, Drie Eikenstraat 655, B-2650 Edegem, Belgium; 3Center for Cell Therapy and Regenerative Medicine, Antwerp University Hospital, Drie Eikenstraat 655, B-2650 Edegem, Belgium; 4Gynaecologic Oncology Unit, Antwerp University Hospital, Drie Eikenstraat 655, B-2650 Edegem, Belgium

**Keywords:** metastatic, recurrent, cervical cancer, PD-1, PD-L1, biomarker, pembrolizumab, immunotherapy, clinical trial

## Abstract

Even though cervical cancer is partly preventable, it still poses a great public health problem throughout the world. Current therapies have vastly improved the clinical outcomes of cervical cancer patients, but progress in new systemic treatment modalities has been slow in the last years. Especially for patients with advanced disease this is discouraging, as their prognosis remains very poor. The pathogen-induced nature, the considerable mutational load, the involvement of genes regulating the immune response, and the high grade of immune infiltration, suggest that immunotherapy might be a promising strategy to treat cervical cancer. In this literature review, we focus on the use of PD-1 blocking therapy in cervical cancer, pembrolizumab in particular, as it is the only approved immunotherapy for this disease. We discuss why it has great clinical potential, how it opens doors for personalized treatment in cervical cancer, and which trials are aiming to expand its clinical use.

## 1. Introduction

To date, cervical cancer (CC) is one of the most preventable malignancies. Population-wide cytological screening and vaccination campaigns against human papilloma virus (HPV) have led to a 75% reduction in mortality of CC over the past 50 years in industrialized countries [1,2]. Future perspectives are promising too. Australia, for example, is projected to reduce its CC cases to fewer than four per 100,000 women by 2035, combining a nationwide HPV vaccination and cytologic screening program, putting them on track to be the first country in the world to eliminate CC as a public health problem [3]. Nevertheless, we must remain cautious not to rely too much on the protective features of these measures. The latest reports show that CC accounts for an estimated 570,000 new cases and 311,000 deaths annually [4]. Thus, despite prevention efforts, CC still ranks as the fourth most prevalent cancer and a leading cause of cancer-related deaths in women worldwide [4]. Moreover, Van Kriekinge et al. estimated that if in 2014 HPV vaccine rates of 70% would have been achieved globally, CC incidence would have decreased, with about 345,000 new cases and mortality with about 178,000 deaths [5]. Even though such a scenario sounds very promising, CC mortality rates would still exceed those of multiple other malignancies in women, like head-and-neck cancer, leukemia, lymphoma, melanoma, and cancer of the brain and central nervous system [6]. Today, we are far from achieving the 70% threshold in industrialized countries, let alone throughout the world. Considering the growing vaccine hesitancy in a considerable part of the population, this margin is not expected to be acquired soon either. Therefore, effective treatments for CC remain highly warranted. 

A small proportion of CCs (<5%) are truly HPV-negative after sensitive HPV assessment. These tumors are more frequently of the non-squamous subtype, diagnosed at advanced stages, show higher prevalence of lymph node metastases, and have an impaired prognosis [7,8]. HPV-negative CCs have distinct molecular characteristics that show similarities with endometrial cancer, while having a significantly higher epithelial-mesenchymal transition mRNA score and a lower frequency of the APOBEC mutagenesis signature compared to their HPV-positive counterparts [9,10]. This implies that these tumors should be considered as a distinct subtype, which is more difficult to screen for (being HPV-negative), cannot be prevented by HPV vaccination and may need a different treatment approach that more resembles the management of endometrial cancer.

Current first-line management of CC consists of surgery, (chemo)radiotherapy for locally advanced disease, and platinum-based chemotherapy with or without the anti-VEGF drug, bevacizumab, for metastatic CC [11,12]. These treatments have reduced the mortality rate of CC, but their effectiveness is now reaching a plateau. Especially for patients with advanced stage CC, this is unfortunate, because their prognosis remains very poor, with a median overall survival (OS) of about 17 months and an estimated 5-year survival of about 17% [13]. In addition, current treatment strategies are generally associated with unwanted adverse events (AEs) and a reduced quality of life for the patient. In recent decades, little progress has been made in the development of better systemic treatments, and early clinical trials with targeted therapies have yet to identify drugs with superior response rates [14]. Therefore, there is an unmet need for innovative therapies to increase durable responses, reduce substantial toxicities associated with current treatment strategies, and improve patients’ lives and outcomes. One such strategy to battle the cancer and improve the long-term benefits of treatment is to stimulate the patient’s own immune system to eradicate residual cancer cells and prevent recurrence by inducing anti-tumor immunity via immunotherapy. In the present review of the literature, we discuss the use of programmed cell death protein 1 (PD-1 or CD279) -targeting immunotherapy for the treatment of recurrent or metastatic CC. We zoom in on the clinical context of pembrolizumab, as this compound is implemented in the second line for the treatment of recurrent or metastatic CC.

## 2. Rationale for PD-1 Blocking Therapy in Cervical Cancer

A sustained HPV infection has a crucial etiological role in most CC [15]. Recently, some of the regulatory networks involved in the carcinogenesis of this disease have been identified, including TGF-beta, C-MYC, MAPK signaling, and APOBEC mutagenesis [9,10]. In a landmark study on invasive CC that was conducted as part of The Cancer Genome Atlas (TCGA) project, amplifications in multiple checkpoint-controlling immune targets have been identified in the tumor cells, such as programmed death ligand 1 (PD-L1, encoded by the *CD274 gene*) and 2 (PD-L2, encoded by *PDCD1LG2*), and in the long non-coding RNA of BRCA4 in the immune cells that regulates the expression of cytosolic immune effector genes, perforin and granzyme A [9]. In addition, CC has been shown to rank amongst the top ten tumors with most somatic mutations, which is associated with a high frequency of neoantigen formation and a better response to immunotherapy [16,17]. CC also ranks among the top ten tumors with the richest tumor immune infiltrate, with CD8^+^ cytotoxic T lymphocytes (CTLs) and macrophages (MΦ) being the most enriched infiltrates, creating a hostile immune environment for the tumor [18,19]. Together, the pathogen-induced nature of the disease (HPV antigens), the considerable mutational load, the strong involvement of genes regulating the immune response, and the relatively high grade of immune infiltration suggests that immunotherapeutic strategies may be promising to treat CC [20,21]. This hypothesis is supported by multiple studies that confirm that clinical responses are seen in CC treated with immune checkpoint blockade (ICB), alone or in combination with other therapies [22].

In recent years, the targeted inhibition of the PD-1 axis for the treatment of cancer has garnered a lot of attention. PD-1 is a co-inhibitory cell-surface receptor, expressed mainly in B- and T-cells, that acts to restrain T-cell-mediated immune responses when activated by its ligands, PD-L1 or PD-L2 [23]. As such, the PD-1 pathway functions as a built-in protection mechanism that tampers the adaptive immune response, thereby preventing immune cell overstimulation and maintaining self-tolerance. Hence, it has an essential role in regulating the balance within the immune system. During persistent antigen encounter, however - like in cancer - PD-1 expression and that of its ligands is often high and sustained and can therefore limit protective immunity in favor of the disease [24]. As a result, CTLs encounter dysfunction and exhaustion within the tumor microenvironment (TME), which leads to adaptive immune resistance. CTLs have long been proven essential in the battle against cancer, so losing these soldiers to cancer-induced immunosuppression could be an important reason why the disease can thrive. Thus, countering this process with PD-1-inhibiting compounds could be a valid strategy to treat cancer. Early clinical trials showed promising results with durable anti-tumor immune responses [25], which led to the approval of multiple PD-1- and PD-L1-targeting monoclonal antibodies for therapeutic use in various cancer types (Table 1). 

In CC, members of the PD-1 axis are upregulated during cancer progression [26]. PD-L1 expression is more apparent in squamous cell carcinoma (34%) than in adenocarcinoma (17%) and adenosquamous carcinoma (29%) [27,28,29,30]. The genes encoding PD-L1 and PD-L2, the *CD274 gene* and *PDCD1LG2*, respectively, were co-amplified or gained extra chromosomes in 67% of CC cases [31]. Several studies found that HPV-positivity is correlated with increased PD-L1 expression [32,33,34]. In the TCGA cohort, PD-L1 methylation was negatively correlated with PD-L1 mRNA expression and associated with HPV infection [35]. This suggests that PD-L1 methylation is a mechanism involved in transcriptional silencing after HPV infection in CC, as also described for other proteins [36]. Moreover, a higher level of PD-1/PD-L1 expression was shown in tumor-infiltrating lymphocytes (TILs) in CC, compared to other tumor types [34]. PD-L1 engagement on T-cells was recently shown to promote the self-tolerance and suppression of neighboring macrophages and effector T-cells in cancer and could predict response to anti-PD-L1 therapy [37,38]. On this basis, together with the immunogenicity of CC, several studies investigated the role of PD-1 blocking therapy in CC. The promising results following the use of the PD-1 blocking antibody, pembrolizumab, in recurrent or metastatic CC, in the KEYNOTE-28 (NCT02054806) and KEYNOTE-158 (NCT02628067) trials, led the Food and Drug Administration (FDA), on 12 June 2018, to approve pembrolizumab for the second-line treatment of PD-L1-positive metastatic or recurrent CC. The European Medicines Agency (EMA), on the other hand, has not yet approved pembrolizumab for the treatment of CC, even in PD-L1-positive tumors.

KEYNOTE-28 (NCT02054806) was a nonrandomized phase IB basket trial of 20 different cohorts in 477 patients with PD-L1-positive advanced solid tumors—including 24 CC cases—that received 10 mg/kg pembrolizumab monotherapy every two weeks for up to 24 months [39]. PD-L1-positivity was assessed using an archived FFPE tumor sample or a newly obtained core or excisional biopsy sample and defined as membranous staining on ≥1% in a modified proportion score or interface pattern as assessed using a laboratory-developed prototype IHC assay with the 22C3 antibody (combined positive score [CPS]). At a median follow-up of 11 months, overall response rate was 17% (95% CI, 5% to 37%). Four patients (17%) achieved a confirmed partial response (PR), and three patients (13%) had stable disease. The six-month progression free survival (PFS) was 13% and six-month OS was 66.7%. The median duration of response for the four patients who achieved a PR was 5.4 months (4.1 to 7.5 months). Treatment-related AEs were experienced by 18 patients (75%): rash (*n* = 2; 21%) and pyrexia (*n* = 4; 17%) were observed and occurred in ≥10% of patients. Five patients experienced grade 3 treatment-related AEs. No grade 4 AEs or deaths were observed. Two patients discontinued treatment because of grade 3 treatment-related AEs (Guillain–Barré syndrome and colitis). Immune-mediated AEs were observed in six patients and included rash (*n* = 2; grade 3), colitis (*n* = 1; grade 3), Guillain–Barré syndrome (*n* = 1; grade 3), hyperthyroidism (*n* = 1; grade 2), and hypothyroidism (*n* = 1; grade 2). These results suggest that, in patients with PD-L1-positive advanced CC, pembrolizumab demonstrates anti-tumor activity and exhibits a safety profile consistent with that seen in other tumor types.

KEYNOTE-158 (NCT02628067) is an ongoing phase II basket trial including 1595 patients with advanced solid tumors, of which the results of 98 patients with previously treated advanced CC have already been published [40]. A total of 82 (83.7%) CC patients had PD-L1-positive tumors (as determined with CPS), of which 77 received one or more lines of chemotherapy. Pembrolizumab 200 mg/kg monotherapy was given every 3 weeks for two years until progression. The primary endpoint was objective response rate on RESIST, assessed by an independent radiological review, and median follow-up was 10.2 months. Objective responses were seen in 12 patients (12.2%), all with a PD-L1-positive tumor (response rate in this group 12/82 = 14.6%), and in 11 out of the 77 patients previously treated with one or more lines of chemotherapy. At the time of interim analysis, the median duration of response was not reached (range >3.7 to 18.6 months). Median PFS was 2.1 months both in the entire population as in the PD-L1 group, and median OS was 9.4 and 11 months, respectively. Treatment-related AEs occurred in 64 (65.3%) patients. The most common were hypothyroidism (10.2%), loss of appetite (9.2%), and fatigue (9.2%). Grade 3 and 4 toxicity were seen in 12 (12.2%) patients (most frequent elevated transaminases 3.1%), which resulted in four patients (4.1%) discontinuing treatment. Immune-related AEs were reported in 25.5% of patients, of which 5.1% were grade 3 or 4 (hepatitis *n* = 2, skin reaction *n* = 2, adrenal insufficiency *n* = 1). The most commonly observed were hyperthyroidism (11.2%) and hypothyroidism (9.2%). No treatment-related deaths occurred. The authors concluded that pembrolizumab monotherapy demonstrated durable anti-tumor activity and manageable safety in patients with advanced CC. Based on these results, the FDA approved the use of pembrolizumab in patients with PD-L1-positive (CPS ≥ 1) advanced CC, progressing during or after chemotherapy.

Different immunotherapy strategies for CC have been extensively investigated and were found to be safe and well-tolerated in early clinical trials [21]. For pembrolizumab, there are no formal contraindications in the clinic and grade 3-4 AEs are uncommon [41]. Most common side effects, including diarrhea, fever, nausea, pain, fatigue, rash, etc., can easily be managed with symptomatic treatments or by dose reduction. On rare occasions, more severe AEs occur, like endocrinopathies, hematological toxicity, severe skin reactions, and immune-related AEs (hepatitis, pneumonitis, nephritis, colitis, etc.). In such cases, pembrolizumab should be temporarily withheld and can be resumed when recovered to grade 1, according to the schemes provided by the manufacturer. Pembrolizumab should be permanently discontinued after any life-threatening adverse reaction and recurrent grade 2 or grade 3–4 immune-related AEs. During pregnancy, the PD-1/PD-L1 pathway was shown to maintain immune tolerance to the fetal allograft; therefore, fetal harm may occur when pembrolizumab is administered to a pregnant woman; however, currently, no human data are available on the risk of embryo-fetal toxicity [42].

Current treatment strategies have reduced the mortality rates of CC but are now reaching a plateau, leading to a stagnation of the treatment progress over the last years. These treatments do not specifically target cancer cells, and are therefore generally associated with unwanted AEs and a reduced quality of life for the patient. Hence, there is an unmet need for innovative therapies to increase durable responses, reduce substantial toxicities associated with current treatment strategies and improve patients’ lives and outcomes. One such strategy that holds promise in all these features is immunotherapy, which has revolutionized the field of oncology in recent years. Benefits include its potential to specifically target cancer cells, to induce systemic anti-tumor immune memory and to mediate long-term survival. Even though the response rates observed in the KEYNOTE-28 and KEYNOTE-158 trials are limited, these features underline the advantages of (anti-PD-1) immunotherapy over the current standard of care for CC. 

## 3. Biomarkers for PD-1 Blocking Therapy in Cervical Cancer

The response rates to ICB across different tumor types emphasize the importance of biomarkers to identify the patients that will benefit, allowing for personalized treatment. PD-L1 protein expression can be used to evaluate the efficacy of immunotherapy, as it is an indicator for immune cell activation via the interferon gamma cascade [43]. Several studies have shown that PD-1 and PD-L1 expression are mainly regulated by interferon gamma signaling via the IL-6/JAK/STAT pathway [44,45,46,47,48,49,50,51]. More importantly, PD-L1 expression can be used as a biomarker to predict the effect of immunotherapy. Scoring for PD-L1 expression is usually performed via immunohistochemistry (IHC) on formalin-fixed paraffin-embedded (FFPE) tissues. For this, two main PD-L1 scoring strategies are used, the CPS and the tumor proportion score (TPS). The CPS is evaluated by IHC with the anti-PD-L1 mouse monoclonal antibody, 22C3 (pharmDx, Agilent DAKO), and is determined by the number of PD-L1 staining cells (tumor cells, lymphocytes, macrophages), divided by the total number of viable tumor cells, multiplied by 100. The TPS is also based on IHC with the 22C3 antibody and is defined as the percentage of viable tumor cells with partial or complete PD-L1 membrane staining at any intensity. Both the CPS and TPS scoring systems are used in the clinic to decide if a patient is suitable for PD-1 targeting therapy [52]. In the KEYNOTE-158 trial, response of CC patients to pembrolizumab was significantly correlated to both the CPS (*p* = 0.008) and TPS (*p* = 0.023), but the CPS identified more responders [40]. Therefore, the CPS is now a validated quantitative scoring system for the detection of PD-L1 on CC FFPE tissues, which determines the eligibility of a CC patient for pembrolizumab treatment, according to the FDA. Based upon the KEYNOTE-28 and KEYNOTE-158 trials, a CPS ≥1 is required before pembrolizumab treatment is justified. Intra- and interobserver concordance of this assay is above 98% in CC [53]. IHC for PD-L1 with 22C3 is a companion assay that is now used in many studies beyond CC to identify patients who may benefit from pembrolizumab [52].

The prognostic role of the expression of members of the PD-1 axis in CC is not very clear and highly context-specific, depending on different factors like the source, transience, and pattern of expression. For example, PD-1 is also expressed on the immunosuppressive regulatory T-cells (Tregs). Thus, PD-L1 expression can support anti-tumor immunity by attenuating the immunosuppressive effects of Tregs. As such, CC patients with a relative excess of infiltrating Tregs show a better survival when the tumor was PD-L1-positive [54]. PD-1 expression on CTLs, on the other hand, is detrimental in CC and might be important to predict the efficacy of PD-1-blocking therapy [55]. Another important prognostic factor is the heterogeneity of the expression, which is difficult to detect. Significantly poorer survival rates were seen in CC patients with diffuse PD-L1 expression compared to patients with marginal PD-L1 expression on the tumor–stroma interface [26]. Another, barely detectable factor is the transience of PD-L1 expression due to fluctuating interferon gamma expression in the TME, which might explain the varying response rates of PD-1 targeting therapy in CC [56]. Real-time screening methods like immuno-Positron Emission Tomography (immune-PET) with radiolabeled antibodies, might better predict response to targeted (immuno)therapies [57]. This technique combines the superior sensitivity of PET imaging with the benefits of the high targeting specificity of monoclonal antibodies. As such, it can provide information on whole-body biomarker distribution or (heterogenic) tumor target expression and act as a companion diagnostic tool in vivo in a non-invasive and longitudinal manner. Currently, multiple immuno-PET imaging techniques are under investigation for PD-L1 biomarker assessment (NCT03746704, NCT04006522, NCT03514719, NCT03065764) [58,59,60].

Recent evidence suggests that a large mutational burden will generate neoantigens for T-cell recognition, leading to the recruitment of CTLs that are mandatory for effective immunotherapy [61]. While a high tumor mutational burden (TMB) has been shown to predict the response to ICB and clinical benefit in some studies [62,63,64,65,66,67], it failed to do so in several others [68,69,70,71]. Ott et al. analyzed PD-L1 expression, the T-cell inflamed gene expression profile (GEP), and non-synonymous tumor mutations assessed by whole exome sequencing (TMB) in patients enrolled in the KEYNOTE-28 trial. PD-L1 expression, a T-cell inflamed GEP, and a high TMB each predicted response to pembrolizumab in multiple tumor types. The correlations between TMB and GEP or PD-L1 were low. However, response patterns indicate that patients with tumors with high levels of both TMB and one of the inflammatory markers (GEP or PD-L1) have the highest probability of responding. Yang et al. developed an immune-related gene (IRG) signature to predict survival and response to immunotherapy in CC patients [72]. They used the TCGA RNA sequencing data to estimate proportions of 22 types of infiltrating immune cells with the CIBERSORT algorithm and downloaded mutation data of 304 CC patients from the TCGA data portal to calculate the TMB. A prognostic IRG signature based on 11 genes was constructed and this proved to be an independent prognostic factor for OS and PFS in CC patients. Seven of those genes were identified as high-risk signatures (*LEPR*, *PRL*, *NRP1*, *TNFRSF10B*, *TNFRSF10A*, *PLAU*, *ANGPTL5*) and four were protective (*PRLHR*, *NR2F2*, *IFI30*, *IGF1*). Based on these signatures, a risk score was established, and the patients were divided into high- and low-risk groups according to the median cutoff of the risk score. In the high-risk group, CTLs and resting mast cells, which were found to be associated with better OS in this study, were lower; and activated mast cells, associated with poorer OS, were higher, compared with the low-risk group. The 11-IRG signature low-risk group represented a more immunogenic phenotype that was more inclined to respond to ICB treatment. In the same study however, the authors could not detect significant differences in TMB and PD-L1 expression between the 11-IRG signature high-risk group and the low-risk group.

Tumor-infiltrating immune subsets might also predict the efficacy of PD-1 targeting therapy. For instance, using triple-color immunofluorescence confocal microscopy, de Vos van Steenwijk et al. could show that a dense infiltration of intraepithelial matured M1-Mϕ and a high CTL/Treg ratio are independent prognostic factors in patients with CC [73]. More recently, the same group identified a CD8^+^FoxP3^+^CD25^+^ T-cell subset as a potential therapeutic target for PD-1-blocking therapy, implicating that this subset may also serve as a predictive biomarker for PD-1-blocking therapy [55].

Ngoi et al. performed a small study specifically focusing on the TME and the molecular genetic profile of tumor samples of four patients with metastatic CC, treated with off-label second line pembrolizumab [74]. All patients received 2 mg/kg pembrolizumab in a 3-weekly regimen upon progression. One patient had a long-lasting PR and remained stable for at least 21 months at the time of reporting the series. The other patients had progressive disease. The responder had a CPS for PD-L1 of 1, and somatic mutations in *ERBB4*, *PIK3CA* and *RB1* were detected.

## 4. Clinical Trials Investigating Pembrolizumab in Patients with Cervical Cancer

Currently there are 28 active recruiting and non-recruiting clinical trials assessing the role of pembrolizumab (combination) therapy in patients with CC. Eleven of those evaluate its therapeutic potential solely in CC patients (Table 2) and are discussed below. In addition, 17 clinical studies investigate the role of pembrolizumab monotherapy, or in combination with chemotherapy, radiotherapy, targeted therapy, immunotherapy, and/or gene therapy in multiple cancer types, including CC (Table 3).

### 4.1. Pembrolizumab Combined with Chemotherapy

The MITO CERV3 trial (NCT04238988) is currently the only trial investigating pembrolizumab combined with chemotherapy solely in patients with CC. It is a single-arm multicenter phase II clinical trial evaluating the role of pembrolizumab in combination with carboplatin–paclitaxel chemotherapy in 45 patients with locally advanced CC. Patients with stage IB2-IIB CC (according to the International Federation of Gynaecology and Obstetrics [FIGO] stage system) will be treated with three cycles of neoadjuvant carboplatin (AUC 5 D1, q21)-paclitaxel (175 mg/mq D1, q21) chemotherapy in combination with pembrolizumab (200 mg flat dose every 3 weeks). After three cycles, non-progressing patients will undergo radical surgery. After surgery, patients presenting with high-risk factors (positive lymph nodes, positive parametria, positive surgical margins or at least two of the following risk factors between tumor diameter >3 cm, lymphovascular space invasion, stromal infiltration >1/3) will receive three cycles of adjuvant carboplatin-paclitaxel with pembrolizumab and maintenance with pembrolizumab 200 mg every 3 weeks until progression, unacceptable toxicity, or patient consent withdrawal for up to 35 cycles.

### 4.2. Pembrolizumab Combined with Targeted Therapy

Currently, the NCT04230954, InnovaTV 205/ENGOT-cx8 (NCT03786081), NCT04641728, and NCT04483544 trials are investigating pembrolizumab in combination with targeted therapy solely in patients with CC. NCT04230954 is a single-arm, open label phase II trial to evaluate the efficacy and safety of cabozantinib (XL184; a small molecule inhibitor of the tyrosine kinases c-Met and VEGFR2, which also inhibits AXL and RET) plus pembrolizumab in 39 patients with recurrent, persistent and/or metastatic CC with PD-L1 tumor positivity. InnovaTV 205/ENGOT-cx8 (NCT03786081) is a seven-arm phase I-II clinical trial investigating the role of tisotumab vedotin (a drug conjugated monoclonal antibody-targeting tissue factor) monotherapy, and in combination with bevacizumab, pembrolizumab, or carboplatin, in 175 subjects with recurrent or stage IVB cervical cancer [75]. The phase I portion of the study is a dose escalation part, whereas the phase II portion is a dose expansion part. Both the NCT04641728 and NCT04483544 trials are single-arm phase II clinical studies aiming to investigate the use of combining pembrolizumab with the PARP inhibitor, Olaparib, in 28 and 48 patients, respectively, with recurrent or metastatic CC. 

### 4.3. Pembrolizumab Combined with Immunotherapy

NCT03444376 and NCT03108495 are both investigating pembrolizumab in combination with other immunotherapy, solely in patients with CC. NCT03444376 is an ongoing single-arm, open-label phase IB-II clinical trial of the combination of GX-188E (a DNA vaccine shown to induce HPV E6- and E7-specific T-cell responses and lesion regression in patients with cervical precancer) with pembrolizumab in 60 patients with advanced, non-resectable HPV-positive CC. Most recently, their interim analysis of 36 patients, of which 26 patients were evaluable for interim activity assessment with at least one post-baseline tumor assessment at week 10, were published in *The Lancet Oncology* [76]. At 24 weeks, 11 (42%; 95% CI 23–63) of 26 patients achieved an overall response; four (15%) had a complete response and seven (27%) had a PR. A total of 16 (44%) of 36 patients had treatment-related AEs of any grade and four (11%) had grade 3–4 treatment-related AEs. Grade-3-increased aspartate aminotransferase, syncope, pericardial effusion, and hyperkalemia, and grade-4-increased alanine aminotransferase were reported in one patient each. No treatment-related deaths were reported. These results suggest that combinatorial treatment of GX-188E with pembrolizumab for patients with recurrent or metastatic CC is safe and showed preliminary anti-tumor activity, which could represent a new potential treatment option for this patient population. NCT03108495 is a five-arm, open-label phase II interventional study evaluating the combination of pembrolizumab and adoptive cell therapy with autologous TIL infusion (LN-145), followed by IL-2, after a non-myeloablative lymphodepletion preparative regimen for the treatment of patients with recurrent, metastatic, or persistent CC [77].

### 4.4. Pembrolizumab Combined with Multiple Other Therapies

NCT02635360, KEYNOTE-826 (NCT03635567), ENGOT-cx11/KEYNOTE-A18 (NCT04221945), and NCT03367871 all investigate pembrolizumab in combination with multiple other therapies, solely in patients with CC. NCT02635360 is an ongoing phase II clinical study to evaluate the safety and effectiveness of pembrolizumab in combination with cisplatin chemotherapy and brachytherapy radiation (chemoradiation) for the treatment of advanced CC [78]. After chemoradiation is complete, 88 subjects will receive pembrolizumab 200 mg IV every 21 days for 3 months during and after the chemoradiotherapy. The KEYNOTE-826 trial is a currently enrolling phase III double-blind randomized multicenter study evaluating the efficacy and tolerability of platinum- and taxane-based chemotherapy with or without pembrolizumab and/or bevacizumab for first-line treatment in patients with persistent, metastatic or recurrent CC [79]. A total of 600 eligible patients will be randomized 1:1 to chemotherapy (paclitaxel 175 mg/m^2^+ cisplatin 50 mg/m^2^ or carboplatin AUC5 with or without bevacizumab 15 mg/kg) + pembrolizumab 200 mg, or placebo every 3 weeks. Patients will be stratified according to metastasis status at diagnosis, planned bevacizumab use (yes or no) and tumor PD-L1 status (CPS <1.1 to <10, or ≥10). Treatment will continue for maximum 35 cycles (about two years), until disease progression, unacceptable toxicity, or voluntary patient withdrawal. The ENGOT-cx11/KEYNOTE-A18 trial is a randomized phase III study of chemotherapy and radiotherapy with pembrolizumab or placebo to pembrolizumab for the treatment of locally advanced CC, aiming to enroll 980 patients [80]. Participants receive placebo or 200 mg of pembrolizumab on Day 1 of each 3-week cycle (Q3W) for five cycles followed by placebo on Day 1 of each 6-week cycle (Q6W) for an additional 15 cycles. During the Q3W dosing period of placebo, participants receive concurrent chemoradiotherapy. The standard of care chemoradiotherapy regimen includes cisplatin 40 mg/m2 IV once per week (QW) for 5 weeks plus 45–50 Gray units (Gy) of external beam radiotherapy given over 40 days, followed by 25–30 Gy of brachytherapy given with the total duration of radiation treatment, not exceeding 56 days. NCT03367871 is a phase II single arm study to evaluate the efficacy of the combination of standard chemotherapy with bevacizumab with Pembrolizumab in women with recurrent, persistent, or metastatic cervical cancer. On day 1 of each 21-day cycle, participants will be administered pembrolizumab 200 mg (IV); chemotherapy including paclitaxel 175 mg/m^2^ or 135 mg/m^2^ (IV), and cisplatin 50 mg/m^2^ (IV) or carboplatin AUC 5; and bevacizumab 15 mg/kg (IV).

## 5. Concluding Remarks

Limited clinical data exist on the use of PD-1-blocking therapy for CC, mainly provided by the KEYNOTE-28, KEYNOTE-158, and NCT03444376 trials. These trials suggest that PD-1 targeting therapy in CC has an acceptable toxicity profile, with fewer AEs than standard treatment, and some promising anti-tumor activity, especially in patients with PD-L1-positive tumors. On this behalf, the FDA, but not the EMA, approved the use of pembrolizumab for the treatment of recurrent or metastatic CC if the tumor shows a CPS of ≥1. Even though CC is immunogenic, only low response rates are seen when the patients are treated with current ICB, including pembrolizumab [20]. Whereas belief in ICB for the treatment of CC is supported, consensus dictates that it currently does not live up to its full potential. Anti-CTLA-4 therapy with ipilimumab could not produce equivalent response rates in CC, as seen with pembrolizumab [82]. The activity of such ICB agents in monotherapy appears to be limited in CC; however, their combinations can elicit synergistic effects [83]. Ongoing efforts are being made to discover biomarkers to stratify for the patients that will benefit from this approach and to identify and target other pathways that might tamper the effect of current ICB therapy [51,84,85,86]. Some interesting parameters still lack current clinical data of pembrolizumab in CC, such as duration of the effects, long-term AEs, and correlation with other predictive biomarker approaches. The latter is an important factor. The FDA approving pembrolizumab based on a predictive biomarker is the first step in personalized treatment for CC, which should pose an example to implement more personalized and precision approaches for CC treatment. Real-time target assessment with immuno-PET imaging using radiolabeled monoclonal antibodies should be considered for this matter.

Standard treatments have had an immense effect on CC patients’ lives and outcome but are now falling short of benefitting the subgroup of patients in the most need of effective treatment. PD-1 targeting immunotherapy poses a potential solution for these patients, but still lacks sufficient effect to be implemented in the first line. Nevertheless, this approach poses a solid basis for further strategies in CC, around which future investigations should work. 

## Figures and Tables

**Table 1 ijms-22-01807-t001:** Current FDA/EMA approved PD-1 axis blockers.

Target	Active Substance	Trade Name	Marketing Holder	FDA Approval	EMA Approval
*PD-1*	Nivolumab	OPDIVO^®^	Bristol-Myers Squibb Pharma EEIG	Since 2014: cHL, ESCC, HCC, HNSCC, Melanoma, MSI-H/dMMR CRC, NSCLC, RCC, SCLC, Urothelial Carcinoma	Since 2015: cHL, HNSCC, Melanoma, NSCLC, RCC, Urothelial Carcinoma
	Pembrolizumab	KEYTRUDA^®^	Merck Sharp & Dohme B.V.	Since 2014: CC, cHL, cSCC, ESCC, Endometrial Carcinoma, Gastric or Esophageal Junction Cancer, HCC, HNSCC, Melanoma, MCC, MSI-H/dMMR Cancer, NSCLC, PMBCL, RCC, SCLC, TMB-H Cancer, TNBC, Urothelial Carcinoma	Since 2015: cHL, HNSCC, Melanoma, NSCLC, RCC, Urothelial Carcinoma
	Cemiplimab	LIBTAYO^®^	Regeneron Ireland U.C.	Since 2018: cSCC	Since 2019: cSCC
*PD-L1*	Atezolizumab	TECENTRIQ^®^	Roche Registration GmbH	Since 2016: HCC, Melanoma, NSCLC, SCLC, TNBC, Urothelial Carcinoma	Since 2017: HCC, NSCLC, SCLC, TNBC, Urothelial Carcinoma
	Avelumab	BAVENCIO^®^	Merck Europe B.V.	Since 2017: MCC, RCC, Urothelial Carcinoma	Since 2018: MCC, RCC, Urothelial Carcinoma
	Durvalumab	IMFINZI^®^	AstraZeneca AB	Since 2017: NSCLC, SCLC, Urothelial Carcinoma	Since 2019: NSCLC

CC = Cervical Cancer; cHL = classical Hodgkin Lymphoma; CRC = Colorectal Cancer; cSCC = cutaneous Squamous Cell Carcinoma; dMMR = deficient MisMatch Repair; EMA = European Medicines Agency; ESCC = Esophageal Squamous Cell Carcinoma; FDA = U.S. Food and Drug Administration; HCC = Hepatocellular Carcinoma; HNSCC = Head and Neck Squamous Cell Carcinoma; MCC = Merkel Cell Carcinoma; MSI-H = MicroSatellite Instability-High; (N)SCLC = (Non-)Small Cell Lung Carcinoma; PMBCL = Primary Mediastinal Large B-Cell Lymphoma; RCC = Renal Cell Carcinoma; TMB-H = Tumor Mutational Burden-High; TNBC = Triple-Negative Breast Cancer; PD-(L)1 = Programmed Death-(Ligand)1.

**Table 2 ijms-22-01807-t002:** Current recruiting and non-recruiting clinical trials assessing pembrolizumab solely in patients with cervical cancer.

NCT	Acronym	Phase	Intervention	Enrollment	Status	Completion	Ref.
**Chemotherapy combinations**							
NCT04238988	MITO CERV 3	II	Neoadjuvant Carbo + Paclitaxel + Pembro	45	Not yet recruiting	September 2023	
**Targeted therapy combinations**							
NCT04230954		II	Cabozantinib + Pembro	39	Recruiting	January 2022	
NCT03786081	InnovaTV 205/ENGOT-cx8	I-II	Tisotumab Vedotin + Pembro	175	Active, not recruiting	April 2022	[75]
NCT04641728		II	Olaparib + Pembro	28	Not yet recruiting	December 2023	
NCT04483544		II	Olaparib + Pembro	48	Recruiting	November 2031	
**Immunotherapy combinations**							
NCT03444376		I-II	GX-188E + Pembro	60	Recruiting	December 2023	[76]
NCT03108495		II	LN-145 + Pembro	138	Recruiting	December 2026	[77]
**Multiple combinations**							
NCT03144466	PAPAYA	I	Cis + RT + BT + Pembro	1	Terminated	January 2019	
NCT02635360		II	Cis + BT + Pembro	88	Recruiting	October 2021	[78]
NCT03635567	KEYNOTE-826	III	Cis + Carbo + Paclitaxel + Bevacizumab + Pembro or Placebo	600	Active, not recruiting	November 2022	[79]
NCT04221945	ENGOT-cx11/KEYNOTE-A18	III	Cis + EBRT + BT + Pembro or Placebo	980	Recruiting	December 2024	[80]
NCT03367871		II	Cis + Carbo + Paclitaxel + Bevacizumab + Pembro	40	Recruiting	October 2025	

BT = Brachytherapy; Carbo = Carboplatin; Cis = Cisplatin; ERBT = External Beam Radiotherapy; Pembro = Pembrolizumab; RT = Radiotherapy.

**Table 3 ijms-22-01807-t003:** Current clinical trials assessing pembrolizumab in multiple malignancies, among which cervical cancer.

NCT	Acronym	Phase	Intervention	Enrollment	Status	Completion	Ref.
**Monotherapy**							
NCT02054806	KEYNOTE-28	I	Pembro	477	Active, not recruiting	December 2023	[39]
NCT02628067	KEYNOTE-158	II	Pembro	1595	Recruiting	June 2026	[40]
NCT03755739		II-III	Pembro	200	Recruiting	November 2033	
**Targeted therapy combinations**							
NCT04432857		I	AN0025 + Pembro	84	Recruiting	March 2023	
NCT04357873	PEVOsq	II	Vorinostat + Pembro	111	Recruiting	December 2024	
NCT03849469	DUET-4	I	XmAb^®^22841 + Pembro	242	Recruiting	March 2027	
**Immunotherapy combinations**							
NCT04099277		I	LY3435151 + Pembro	2	Terminated	March 2020	
NCT03277352		I-II	INCAGN01876 + Epacadostat + Pembro	10	Completed	July 2020	
NCT03228667	QUILT-3.055	II	N-803 + Pembro +/- PD-L1 t-haNK	636	Recruiting	August 2021	
NCT04234113		I	SO-C101 + Pembro	96	Recruiting	March 2022	
NCT03841110		I	FT500 + Pembro +/- IL-2	76	Recruiting	June 2022	
NCT03799003		I	ASP1951 + Pembro	435	Recruiting	October 2023	
NCT03454451		I	CPI-006 + Pembro	378	Recruiting	December 2023	[81]
NCT03311334		I-II	DSP-7888 + Pembro	104	Recruiting	February 2024	
**Gene therapy combinations**							
NCT03544723		II	Ad-p53 + ICI, among which Pembro	40	Recruiting	December 2022	
**Multiple combinations**							
NCT03192059	PRIMMO	II	RT + Vit D + Aspirin + Lansoprazole + Cyclophosphamide + Curcumin + Pembro	43	Recruiting	June 2022	
NCT04116320	AM-003	I	Echopulse + Imiquimod + Pembro	32	Recruiting	May 2023	
NCT04652076	GYNET	I-II	NP137+ Cis + Carbo + Pembro	240	Not yet recruiting	November 2024	

Carbo = Carboplatin; Cis = Cisplatin; ICI = Immune Checkpoint Inhibitor; Pembro = Pembrolizumab; RT = Radiotherapy.

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
