# Peer review of "Targeting the PD-1 Axis with Pembrolizumab for Recurrent or Metastatic Cancer of the Uterine Cervix: A Brief Update"

_ijms, 2021, doi:10.3390/ijms22041807_

Round 1

Reviewer 1 Report

In this manuscript, the authors discuss the usage of immunotherapy for cervical cancer treatment. They provide a comprehensive overview in this particular topic and provide some information about the current clinical trial with anti-PDL1/2 treatment. This content in this review is suitable for the readers to get general information and a quick overview about using anti-PDL1/2 in cervical cancer treatment. Overall, this manuscript is recommended especially for those whose are new in this area.

Some suggestions:

  1. The response rate of pembrolizumab is a bit low mentioned in KEYNOTE-28 and KEYNOTE-158. This means not much PD-L1 positive patients will be benefited. The authors may discuss the advantages of using pembrolizumab over the traditional treatments. So that the readers will have a more complete understanding on anti-PD-L1 treatment.
  2. The authors may mention if there was reported severe adverse effect after pembrolizumab treatment. Also, the authors can discuss the condition in which the PD-L1 positive patients should avoid from pembrolizumab.
  3. In section 2, line 3, the authors mention “C-MYK”. Is it C-MYC?

Author Response

  1. The response rate of pembrolizumab is a bit low mentioned in KEYNOTE-28 and KEYNOTE-158.
    This means not much PD-L1 positive patients will be benefited. The authors may discuss the advantages of using pembrolizumab over the traditional treatments. So that the readers will have a more complete understanding on anti-PD-L1 treatment.
    A new paragraph was added to section 2, explaining the advantages of anti-PD-1 therapy over the current standard of care for CC. In addition, some content was added to the final paragraph of section
    1 to introduce the readers to how the current standard of care could be improved with (anti-PD-1) immunotherapy.
    2. The authors may mention if there was reported severe adverse effect after pembrolizumab treatment.
    Also, the authors can discuss the condition in which the PD-L1 positive patients should avoid from pembrolizumab.
    Adverse events and conditions to avoid pembrolizumab are now discussed in a new paragraph in section 2 of the revised manuscript.
    3. In section 2, line 3, the authors mention “C-MYK”. Is it C-MYC?
  2. ‘C-MYK’ was corrected into ‘C-MYC’ in the revised manuscript, thank you for noticing.

Reviewer 2 Report

Reviewer's comments to « Targeting the PD-1 axis for recurrent or metastatic cancer of the uterine cervix: a brief update” by Verhoeven et al. (IJMS # 1078374).

In this review Authors discuss the role of immune checkpoint inhibition in the therapy of recurrent or metastatic uterine cervix carcinoma. Discussion in focused almost exclusively on pembrolizumab, an anti-PD-1 antibody, as this molecule is currently in use as a second line treatment for metastatic of this pathology. Despite screening and HPV immunization that are projected to greatly decrease the incidence of cervical cancer on a longer term, inoperable disease currently remains an important burden with several hundreds of thousands of deaths per year. Considering the success of immune checkpoint inhibition in the therapy of a selected set of neoplasia such as metastatic melanoma, non-small cell lung cancer and renal carcinoma, and because this type of therapy is in rapid progress in other types of tumors as well, this paper is timely and will interest a wide range of readers involved in the treatment of gynecological malignancies, as well as in research aimed at the improvement of immune checkpoint therapy in general.

After a brief introduction on cervical cancer etiology, epidemiology and current therapy, Authors present the cell biological basis of PD-1-based immune checkpoint therapy, and discuss its side effects and the difficulties related to the role of PD-1 immunohistochemistry and other biomarkers for the prediction of the efficacy of anti-PD-1 antibody treatment. Authors thereafter discuss the clinical use of various anti-PD-1/PD-L1 antibodies already approved for various malignancies or currently in clinical trials, with special attention to cervical cancer, and conclude that pembrolizulab has a potentially interesting antitumor effect also in this pathology. Identification of predictive biomarkers, as well as current clinical trials on cervical cancer and pembrolizumab alone or in combination with other treatment modalities are discussed in detail.  Authors then conclude that immune checkpoint inhibition, and pembrolizumab in particular may prove to be useful in the treatment of advanced or metastatic cervical cancer, pending identification of predictive biomarkers, efficient combination therapy modalities and techniques that will further enhance immunogenicity of tumors.

The Manuscript is well written, and is structured in a clear and comprehensible manner. The current efforts of clinical anticancer research in the field of cervical cancer immunotherapy are well illustrated and discussed. The overview of immune checkpoint inhibition and the discussion of the various clinical trials on pembrolizumab alone or in combination will interest a wide range of readers such as clinicians involved in the treatment of cervical carcinoma, as well as tumor immunologists and people working on cancer combination therapy in general.

Minor comments:

Lanes 29-31: “Population-wide cytological screening and vaccination campaigns against human papilloma virus (HPV) have led to a stumbling 75% reduction in incidence and mortality of CC over the past 50 years in industrialized countries.” In the Reviewer’s opinion the expression “stumbling” is problematic. “Incidence” is stumbling, not “reduction in incidence” (which is in fact increasing).

Some cervical carcinomas are not HPV-related. Could Authors briefly comment on this ?

Lane 52: Please state mechanism of action for bevacizumab (angiogenesis inhibition). Because this paper deals with therapeutic antibodies of the checkpoint inhibitor type, this would bring some additional clarity to readers (i.e.: bevacizumab is not an immune checkpoint inhibitor.)

Lanes 64-65: Authors justify limiting the scope of this paper to pembrolizumab, because this antibody is currently approved for advanced cervical carcinoma. This is quite acceptable. However, for readers not familiar with the subject, it would be useful to briefly discuss other similar antibodies, such as, for example, nivolumab, as well as anti-PD-L1 and CTLA-4-targeted therapy, as these may prove to have some interest for combination therapies with pembrolizumab, potentially.

Lane 69: “APOBEK” : APOBEC ?

Lanes 71-74: “…amplifications in multiple checkpoint-controlling immune targets have been identified, such as programmed death ligand 1 (PD-L1, encoded by the CD274 gene) and 2 (PD-L2, encoded by PDCD1LG2), and the long non-coding RNA of BRCA4, which regulates the expression of cytosolic immune effector genes, perforin and granzyme A.” It would be appropriate here to more clearly separate concepts related to phenomena occurring within cancer cells (PD-L1/2-related) from those occurring in immune cells (perforin, granzyme-related etc.)

Tables should be reformatted wherever possible in order to avoid unnecessary hyphenation of drug names.

Lanes 107-108: “(Non-)Small Cell…” : “Non-Small Cell…” ? Please delete ")".

Lanes 119-120:  “Moreover, a higher level of PD-1/PD-L1 expression was shown in tumor infiltrating lymphocytes (TILs)…”  This phrase requires some clarification/more details, as PD-L1 is usually expected to be expressed on tumor cells, rather than on (effector) lymphocytes.

Lanes 177-178: “PD-L1 protein expression can be used to evaluate the efficacy of immunotherapy, as it is an indicator for immune cell activation via the interferon gamma cascade.” This phrase lack mechanistic detail. In general, a paragraph on the intracellular/biochemical mechanisms of checkpoint inhibition (such as, for example, tyrosine phosphatase activation in immune cells, or mechanisms of regulation of PD-L1/2 expression in tumor cells) would be appropriate in the Introduction.

Lane 195: “Intra- and interobserver variability of this assay is above 98% in CC” This phrase is unclear. If variability is as high as almost 100%, what is the value of the assay ?

Lanes 99-100: What does “transience” mean in this context ? Does this mean that marker expression changes in time ? If there is temporal variability in marker expression, this should be briefly discussed.  

Lane 263: “radiotherapy”

Lanes 390-391: The Acknowledgement section is incomplete.

Author Response

  1. Lanes 29-31: “Population-wide cytological screening and vaccination campaigns against human papilloma virus (HPV) have led to a stumbling 75% reduction in incidence and mortality of CC over the past 50 years in industrialized countries.” In the Reviewer’s opinion the expression “stumbling” is problematic. “Incidence” is stumbling, not “reduction in incidence” (which is in fact increasing).
  2. The sentence has been corrected as suggested by the reviewer.
    2. Some cervical carcinomas are not HPV-related. Could Authors briefly comment on this ?
    A paragraph on HPV-negative CC was added to section 1 of the revised manuscript.
    3. Lane 52: Please state mechanism of action for bevacizumab (angiogenesis inhibition). Because this
    paper deals with therapeutic antibodies of the checkpoint inhibitor type, this would bring some additional
    clarity to readers (i.e.: bevacizumab is not an immune checkpoint inhibitor.)
    It has now been made clear that bevacizumab is an anti-VEGF drug.
    4. Lanes 64-65: Authors justify limiting the scope of this paper to pembrolizumab, because this antibody
    is currently approved for advanced cervical carcinoma. This is quite acceptable. However, for readers
    not familiar with the subject, it would be useful to briefly discuss other similar antibodies, such as, for
    example, nivolumab, as well as anti-PD-L1 and CTLA-4-targeted therapy, as these may prove to have
    some interest for combination therapies with pembrolizumab, potentially.
    Other immune checkpoint antibodies for CC and its use in combinatorial strategies is now addressed in
    section 5.

5. Lane 69: “APOBEK” : APOBEC ?
‘APOBEK’ was corrected into ‘APOBEC’ in the revised manuscript.
6. Lanes 71-74: “…amplifications in multiple checkpoint-controlling immune targets have been identified,
such as programmed death ligand 1 (PD-L1, encoded by the CD274 gene) and 2 (PD-L2, encoded by
PDCD1LG2), and the long non-coding RNA of BRCA4, which regulates the expression of cytosolic
immune effector genes, perforin and granzyme A.” It would be appropriate here to more clearly separate
concepts related to phenomena occurring within cancer cells (PD-L1/2-related) from those occurring in
immune cells (perforin, granzyme-related etc.)
This section was rephrased to make a clearer distinction between the events that occur in the tumor
cells from those in the immune cells.
7. Tables should be reformatted wherever possible in order to avoid unnecessary hyphenation of drug
names.
Tables were reformatted as suggested.
8. Lanes 107-108: “(Non-)Small Cell…” : “Non-Small Cell…” ? Please delete ")".
With the brackets in (N)SCLC we address both SCLC and NSCLC in one explanation, in analogy with
‘PD-(L)1 = Programmed Death-(Ligand)1’ in the same paragraph.
9. Lanes 119-120: “Moreover, a higher level of PD-1/PD-L1 expression was shown in tumor infiltrating
lymphocytes (TILs)…” This phrase requires some clarification/more details, as PD-L1 is usually
expected to be expressed on tumor cells, rather than on (effector) lymphocytes.
PD-L1 engagement on CD8+ T-cells is a known phenomenon that could even predict anti-PD-L1
therapy. A sentence was added to the revised manuscript along with the appropriate references to
support this statement.
10. Lanes 177-178: “PD-L1 protein expression can be used to evaluate the efficacy of immunotherapy,
as it is an indicator for immune cell activation via the interferon gamma cascade.” This phrase lack
mechanistic detail. In general, a paragraph on the intracellular/biochemical mechanisms of checkpoint
inhibition (such as, for example, tyrosine phosphatase activation in immune cells, or mechanisms of
regulation of PD-L1/2 expression in tumor cells) would be appropriate in the Introduction.
We have added a sentence supported by multiple appropriate references to account for the underlying
mechanisms of this phenomenon.
11. Lane 195: “Intra- and interobserver variability of this assay is above 98% in CC” This phrase is
unclear. If variability is as high as almost 100%, what is the value of the assay ?
This phrase was corrected: ‘variability’ was replaced by ‘concordance’.
12. Lanes 99-100: What does “transience” mean in this context ? Does this mean that marker expression
changes in time ? If there is temporal variability in marker expression, this should be briefly discussed.
Transience of PD-L1 expression is now explained in the revised manuscript.
13. Lane 263: “radiotherapy”
‘Radiotherap’ is now corrected to ‘radiotherapy’, thank you for noticing.
14. Lanes 390-391: The Acknowledgement section is incomplete.
Acknowledgements are now completed in the revised manuscript. Thank you again for the valuable
suggestions.

Reviewer 3 Report

1st sentence in abstract, cancer is highly preventable! is a very misleading statement and should be removed. The review should be focused on accumulating published fact and making a conclusion out of it. Are all therapies reaching a plateau? Authors should refrain from making such statements. 

Overall the claims made for immunotherapy to be a promising intervention for cervical cancer should be systematically addressed and proved with adequate reference. Theres needs to be title wise organisation. For example: how come pathogen nature of cancer will effect outcomes from an immunotherapy?

Needs detailed explanation on why HPV vaccine is not being effective against certain type of CC and need for a therapy. 

Line 68 should read; have been identified (not exposed). Exposed is used when some ones hiding something and you find it out. Proper terminology and adjectives should be used throughout the paper, leading to a good flow for reading. 

Title should include the name of pembrolizumab as the paper is focused on it.

Some unnecessary details should be summarised in few words. Try to make the data more concise.  

Author Response

1. 1st sentence in abstract, cancer is highly preventable! is a very misleading statement and should be
removed. The review should be focused on accumulating published fact and making a conclusion out
of it. Are all therapies reaching a plateau? Authors should refrain from making such statements.

The abstract has been adapted accordingly.
2. Overall the claims made for immunotherapy to be a promising intervention for cervical cancer should
be systematically addressed and proved with adequate reference. Theres needs to be title wise
organisation. For example: how come pathogen nature of cancer will effect outcomes from an
immunotherapy?

The claims regarding the rationale for immunotherapy in CC have now been clarified in section 2 of the
revised manuscript.
3. Needs detailed explanation on why HPV vaccine is not being effective against certain type of CC and
need for a therapy.

A paragraph on HPV-negative CC was included in section 1.
4. Line 68 should read; have been identified (not exposed). Exposed is used when some ones hiding
something and you find it out. Proper terminology and adjectives should be used throughout the paper,
leading to a good flow for reading.

As suggested by the reviewer, ‘exposed’ was corrected to ‘indentified’.
5. Title should include the name of pembrolizumab as the paper is focused on it.

As suggested by the reviewer, ‘pembrolizumab’ was included in the title of the paper.
6. Some unnecessary details should be summarised in few words. Try to make the data more concise.

An effort was made to make the data more concise. Thank you again for the valuable suggestions.